# The Spectrum of Teleparallel Gravity

**Tomi Koivisto [1,2,3,4,*] and Georgios Tsimperis [5]**

[1]   Nordita, KTH Royal Institute of Technology and Stockholm University, Roslagstullsbacken 23,
      SE-10691 Stockholm, Sweden
[2]   Laboratory of Theoretical Physics, Institute of Physics, University of Tartu, W. Ostwaldi 1,
      50411 Tartu, Estonia
[3]   National Institute of Chemical Physics and Biophysics, Rävala pst. 10, 10143 Tallinn, Estonia
[4]   Helsinki Institute of Physics, P.O. Box 64, FIN-00014 Helsinki, Finland and Department of Physical Sciences,
      Helsinki University, P.O. Box 64, FIN-00014 Helsinki, Finland
[5]   Department of Physics, Stockholm University, AlbaNova University Centre, SE-106 91 Stockholm, Sweden;
      gtrsimp@gmail.com
*   Correspondence: tomi.koivisto@nordita.org

**Abstract:** The observer's frame is the more elementary description of the gravitational field than the metric. The most general covariant, even-parity quadratic form for the frame field in arbitrary dimension generalises the New General Relativity by nine functions of the d'Alembertian operator. The degrees of freedom are clarified by a covariant derivation of the propagator. The consistent and viable models can incorporate an ultra-violet completion of the gravity theory, an additional polarisation of the gravitational wave, and the dynamics of a magnetic scalar potential.

**Keywords:** teleparallel theory of gravity; nonlocal theories of gravity

## 1. Introduction

Gravitational waves, a prediction of General Relativity (GR) that was only recently directly confirmed in the experimental data [1], are most often discussed in terms of fluctuations of the metric field. However, at a more fundamental level, the gravitational field in GR has to be understood as the *frame field*. It has been established long ago that the frame field, in four dimensions a.k.a. the *vierbein*, or the *tetrad*, is necessary for the consistent gravitational coupling of the electron [2]. Moreover[1], the frame field formulation of GR facilitates the covariant definition of gravitational energy-momentum complex [8].

A frame field is a set of $n$ orthonormal vector fields $\{\partial_a\}_{a=0,1,...,n-1}$ defined on an $n$-dimensional Lorentzian manifold that is interpreted as a model of spacetime. All tensorial quantities on the manifold can be expressed using the frame field $\partial_a$ and its dual coframe field $\mathbf{e}^a$. In particular, the components of the contravariant metric tensor, $g^{\mu\nu}$, are obtained from the components $\partial_a{}^\mu$ of the frame field $\partial_a = \partial_a{}^\mu \partial_\mu$ using the Cartan-Killing form $\eta^{ab}$ that is interpreted as the Minkowski metric in the tangent space, as $g^{\mu\nu} = \eta^{ab}\partial_a{}^\mu\partial_b{}^\nu$. The generic frame field has $n^2$ independent components, and it gives rise to the $\frac{1}{2}n(n+1)$ independent components of the symmetric rank-2 tensor $g_{\mu\nu}$. In the case of GR, the remaining $\frac{1}{2}n(n-1)$ components are eliminated by Lorentz invariance.

It is of interest to consider more general theories that may be formulated in terms of the frame field. During the past hundred years, a plethora of alternatives and extensions to GR have been

---

[1]   The teleparallel theory of the frame field has been considered as the gauge theory of the group of translations [3–5]. The main difficulties of this interpretation (that the connection is not generated by translations, nor minimally coupled to matter) are resolved in the recently "purified gravity" [6] which is not however considered in this paper, but see [7].

introduced, and such are currently under extensive investigation especially motivated by the problems of modern cosmology [9]. One of the first things to check in a new theory of gravity is its inherent consistency and observational viability in the limit of Minkowski space. Our aim is to classify the possible theories for the frame field in this limit by the properties of the propagator. The poles of the propagator determine the particle content of the theory.

We shall study the most general frame field action that is Poincaré and parity invariant. The main conclusion will be that there exist viable frame field theories which are not captured by the generic metric theory either in Riemannian [10] or non-Riemannian [7] geometry. When restricting to the second order in derivatives, the action reduces to the well-known Møller-Pellegrini-Plebański theory [8,11] a.k.a. the New GR [12], whose linearisation has been often considered previously [13–20]. The most general action, at the relevant quadratic limit, extends the three-parameter case of New GR by nine functions of the covariant derivative operator, and can accommodate also the ghost- and singularity-free structure that has been previously realised in the metric theories [7,10]. Besides the spin-2 graviton with the infinite-derivative structure and the spin-0 dilaton-like particle that are expected in the closed string field theory, the spectrum of a consistent frame field theory may also feature the spin-0 Kalb-Ramond-like particle.

In the remainder of this brief paper, we shall report the most general quadratic action, Equation (2), and the field Equations (16) and (17) in Section 2, the propagator, Equation (31), in Section 3, and then present our conclusions in Sections 4 and 5.

## 2. Field Equations

The motivation is to uncover the possible properties of gravitation that may not be described by solely the metric. We are interested in the most general theory for the (co)frame field $e^a{}_\mu$, but the properly invariant formulation [21] also includes the spin connection $\omega^a{}_b$, though it is purely inertial [5,21,22]. The field strength of the coframe field is written in the differential form notation as $\mathbf{T}^a = \mathrm{D}e^a = \mathrm{d}e^a + \omega^a{}_b \wedge e^b$, but we shall be explicit with the components such as the $T^a{}_{\mu\nu}$ in

$$\mathbf{T}^a = \frac{1}{2}T^a{}_{\mu\nu}\mathrm{d}x^\mu \wedge \mathrm{d}x^\nu = \left(\partial_{[\mu}e^a{}_{\nu]} + \omega^a{}_{b[\mu}e^b{}_{\nu]}\right)\mathrm{d}x^\mu \wedge \mathrm{d}x^\nu\,. \tag{1}$$

The most general theory that is quadratic in this field strength can be parameterised by nine independent functions of the d'Alembertian operator $\Box = g^{\mu\nu}\nabla_\mu\nabla_\nu$. We write action for the theory as

$$I = -\int \mathrm{d}^n x e L + I_{(M)}\,, \tag{2}$$

where $e = \det e^a{}_\mu = \sqrt{\det g_{\mu\nu}}$, $I_{(M)}$ is the action for the matter fields, and the gravitational Lagrangian is

$$\begin{aligned}
L &= T^{\alpha\mu\nu}\Big\{c_1(\Box)T_{\alpha\mu\nu} + c_2(\Box)T_{\nu\mu\alpha} + g_{\alpha\nu}c_3(\Box)T_\mu \\
&\quad + \Box^{-1}\Big[c_4(\Box)\nabla_\alpha\nabla^\beta T_{\beta\mu\nu} + c_5(\Box)\nabla_\alpha\nabla^\beta T_{\mu\nu\beta} + c_6(\Box)\nabla_\alpha\nabla^\beta T_{\mu\nu\beta} + c_7(\Box)\nabla_\nu\nabla^\beta T_{\mu\alpha\beta} + g_{\alpha\nu}c_8(\Box)\nabla_\mu\nabla^\beta T_\beta\Big] \\
&\quad + \Box^{-2}c_9(\Box)\nabla_\alpha\nabla_\mu\nabla^\rho\nabla^\sigma T_{\rho\sigma\nu}\Big\}\,.
\end{aligned} \tag{3}$$

We have defined the trace $T_\mu = T^\alpha{}_{\mu\alpha}$. This action reduces to the three-parameter New GR [12] when $c_1(\Box) = c_1$, $c_2(\Box) = c_2$ and $c_3(\Box) = c_3$ are constants, and the rest of the functions are zero. Assuming the quadratic torsion is modulated by analytic functions, the five terms in the second line are at least fourth derivative and the one term in the third line are at least sixth order derivative[2].

---

[2]  The inverse operators are included only for the convenience of keeping the $c_i$ dimensionless. Thus, we assume that $c_1$, $c_2$ and $c_3$ are analytic, as well as $c_i/\Box$, where $i = 4, 5, 6, 7, 8$, and $c_9/\Box^2$. Therefore all the apparent inverse d'Alembertians actually cancel from the action and the field equations as well.

The pure-gauge connection $\omega^a{}_b$ is given by a Lorentz transformation $\Lambda^a{}_b$ of the Weitzenböck connection ($\omega^a{}_b = 0$) as $\omega^a{}_b = (\Lambda^{-1})^a{}_c d\Lambda^c{}_b$. The spin connection $\omega_{ab}$ is antisymmetric. Thus, the action principle (2) is understood as $L = L(\mathbf{e}^a, \Lambda^a{}_b)$ [21]. We expand the connection as

$$\omega^a{}_{b\mu} = \partial_\mu A^a{}_b + \mathcal{O}(A^2), \quad \Lambda^a{}_b \approx \delta^a{}_b - A^a{}_b + \mathcal{O}(A^2), \quad \text{where} \quad A_{ab} = A_{[ab]}, \tag{4}$$

and the coframe field as

$$\mathbf{e}^a{}_\mu = \delta^a{}_\mu + B^a{}_\mu, \tag{5}$$

which implies for the inverse

$$\mathbf{\partial}_a{}^\mu = \delta_a{}^\mu - B^\mu{}_a + \mathcal{O}(B^2), \quad B^\mu{}_a = \delta_b{}^\mu \delta_a{}^\nu B^b{}_\nu. \tag{6}$$

At the lowest order, the metric perturbation is given by the symmetric part,

$$g_{\mu\nu} \equiv \eta_{ab} \mathbf{e}^a{}_\mu \mathbf{e}^b{}_\nu = \eta_{\mu\nu} + h_{\mu\nu} + \mathcal{O}(B^2), \quad h_{\mu\nu} \equiv 2B_{(\mu\nu)}, \quad B_{\mu\nu} \equiv \delta_{a\mu} B^a{}_\nu, \tag{7}$$

and for the antisymmetric part we define invariant combination

$$b_{\mu\nu} \equiv 2\left(B_{[\mu\nu]} - A_{[\mu\nu]}\right), \tag{8}$$

in terms of the antisymmetric perturbations of the frame field, $B_{\mu\nu}$, and of the pure-gauge field, $A_{\mu\nu} = \delta_{a\mu} \delta^b_\nu A^a{}_b$. It is then straightforward though tedious to expand the action (2) to the second order in the perturbations. Describing the matter action $I_M$ with the linear source term $\tau^{\mu\nu}$, we obtain

$$I = -\frac{1}{4} \int d^n x \left(L_{(h^2)} + L_{(hb)} + L_{(b^2)}\right) + 2\int d^n x\, B_{\mu\nu} \tau^{\mu\nu} + \mathcal{O}(B^3), \tag{9}$$

where the purely metric part is (we denote the trace $h = \eta^{\mu\nu} h_{\mu\nu}$)

$$L_{(h^2)} = h^{\mu\nu}\left[a(\square)\square h_{\mu\nu} + 2b(\square)\partial^\alpha \partial_\mu h_{\nu\alpha} + c(\square)\left(\partial_\mu \partial_\nu h + \eta_{\mu\nu}\partial_\alpha \partial_\beta h^{\alpha\beta}\right) + \eta_{\mu\nu}d(\square)\square h + \frac{f(\square)}{\square}\partial_\mu \partial_\nu \partial_\alpha \partial_\beta h^{\alpha\beta}\right], \tag{10}$$

there appears the one possible interaction term

$$L_{(hb)} = -2h^{\mu\nu} x(\square)\partial_\mu \partial^\alpha b_{\alpha\nu}, \tag{11}$$

and the part involving only the antisymmetric perturbation is

$$L_{(b^2)} = b^{\mu\nu}\left[y(\square)\square b_{\mu\nu} + 2z(\square)\partial_\mu \partial^\alpha b_{\alpha\nu}\right]. \tag{12}$$

The functions in (10) read (omitting the arguments of $\square$ from now on)

$$a = 2c_1 + c_2 - c_6 - c_7, \tag{13a}$$

$$b = -c_1 - \frac{1}{2}(c_2 - c_3 + c_4 - c_5 - c_6 - 2c_7 + c_9), \tag{13b}$$

$$c = -c_3 + c_8, \tag{13c}$$

$$d = c_3 - c_8, \tag{13d}$$

$$f = c_4 - c_5 - c_7 - c_8 + c_9. \tag{13e}$$

The rest of the functions are specified as

$$x = 2c_1 + c_2 + c_3 - c_4 + c_5 - c_6 - c_9, \tag{14}$$

and

$$y = -c_1 + \frac{3}{2}c_2 + \frac{1}{2}(c_3 - c_4 + c_5 + c_6) + c_7 - \frac{1}{2}c_9, \tag{15a}$$

$$z = 2c_1 - c_2 - c_6 - c_7. \tag{15b}$$

The field equations for the symmetric part, including a source term, are

$$-2\tau_{(\mu\nu)} = a\Box h_{\mu\nu} + 2b\partial^\alpha\partial_{(\mu}h_{\nu)\alpha} + c\left(\partial_\mu\partial_\nu h + \eta_{\mu\nu}\partial_\alpha\partial_\beta h^{\alpha\beta} - \eta_{\mu\nu}\Box h\right) + \frac{f}{\Box}\partial_\mu\partial_\nu\partial_\alpha\partial_\beta h^{\alpha\beta} - x\partial_{(\mu}\partial^\alpha b_{\nu)\alpha}, \tag{16}$$

and the antisymmetric components of the field equations are

$$2\tau_{[\mu\nu]} = y\Box b_{\mu\nu} - 2z\partial_{[\mu}\partial^\alpha b_{\nu]\alpha} + x\partial_{[\mu}\partial^\alpha h_{\nu]\alpha}. \tag{17}$$

The divergence of the symmetric source becomes

$$-2\eta^{\mu\rho}\partial_\mu\tau_{(\rho\nu)} = (a+b)\Box h^\mu{}_{\nu,\mu} + (b+c+f)h^{\alpha\beta}{}_{,\alpha\beta\nu} - \frac{1}{2}x\Box\partial^\mu b_{\mu\nu} = \frac{1}{2}x\left(\Box h^\mu{}_{\nu,\mu} - h^{\alpha\beta}{}_{,\alpha\beta\nu} - \Box\partial^\mu b_{\mu\nu}\right). \tag{18}$$

In the second equality we have taken into account the relations $a + b = -(b + c + f) = x/2$ that follow identically from the definitions (13) and (14). Thus, if the coupling of the $h_{\mu\nu}$ and the $b_{\mu\nu}$ vanishes, $x = 0$, the usual covariant conservation of energy-momentum is recovered. The divergence of the antisymmetric source is

$$-2\eta^{\mu\rho}\partial_\mu\tau_{[\rho\nu]} = (y+z)\Box\partial^\mu b_{\mu\nu} - \frac{1}{2}x\left(\Box h^\mu{}_{\nu,\mu} - h^{\alpha\beta}{}_{,\alpha\beta\nu}\right) = \frac{1}{2}x\left(\Box\partial^\mu b_{\mu\nu} - \Box h^\mu{}_{\nu,\mu} + h^{\alpha\beta}{}_{,\alpha\beta\nu}\right). \tag{19}$$

In the second equality, we have used that $y + z = x/2$, as dictated by the coefficients (14) and (15). Combining the two divergences shows that, to the linear order in perturbations, we have simply $\partial_\mu\tau^\mu{}_\nu = 0$. Another consistency check is that the connection equations of motion are redundant with the equations of motion for the antisymmetric frame field perturbation.

## 3. Propagator

The field $\tilde{B}_{\mu\nu} \equiv B_{\mu\nu} - A_{\mu\nu}$ decomposes into the spin parts

$$\tilde{B}_{\mu\nu} = (2^+) \oplus (1^+) \oplus (1^-) \oplus (1^-) \oplus (0^+) \oplus (0^+) \equiv (\mathbf{g}) \oplus (\mathbf{b}) \oplus (\mathbf{m}) \oplus (\mathbf{e}) \oplus (\mathbf{s}) \oplus (\mathbf{w}). \tag{20}$$

Thus, in terms of the irreducible representations of the rotational group, a rank-2 tensor consists of a tensor piece, one vector and two pseudovector pieces, and two scalars. Along the lines of Refs [13,23], they could be referred to as "gravity", "magnetic", "momentum", "electric", "stress" and "work", respectively. To construct the spin projection operators [13,14,16,23] into the respective subspaces, we define, in terms of the wavevector $k^\mu$, the two bases

$$\theta_{\mu\nu} = \eta_{\mu\nu} - k_\mu k_\nu/k^2, \quad \sigma_{\mu\nu} = k_\mu k_\nu/k^2. \tag{21}$$

The projection operators we need for the symmetric sector can then be defined as

$$P^{(\mathbf{g})}_{\mu\nu\rho\sigma} = \theta_{\mu(\rho}\theta_{\sigma)\nu} - \frac{1}{n-1}\theta_{\mu\nu}\theta_{\rho\sigma}, \tag{22}$$

$$P^{(\mathbf{m})}_{\mu\nu\rho\sigma} = \theta_{\mu(\rho}\sigma_{\sigma)\nu} + \theta_{\nu(\rho}\sigma_{\sigma)\mu}, \tag{23}$$

$$P^{(\mathbf{s})}_{\mu\nu\rho\sigma} = \frac{1}{n-1}\theta_{\mu\nu}\theta_{\rho\sigma}. \tag{24}$$

It turns out that we need only one scalar projector, since the (**w**)-subspace is empty in any possible pure-torsion theory. When taking into account the antisymmetric sector, the following operators need to be introduced.

$$P^{(\mathbf{e})}_{\mu\nu\rho\sigma} = \theta_{\mu[\rho}\sigma_{\sigma]\nu} - \theta_{\nu[\rho}\sigma_{\sigma]\mu} \,, \tag{25}$$

$$P^{(\mathbf{b})}_{\mu\nu\rho\sigma} = \theta_{\mu[\rho}\theta_{\sigma]\nu} \,, \tag{26}$$

$$P^{(\mathbf{m}\times\mathbf{e})}_{\mu\nu\rho\sigma} = \theta_{\mu[\rho}\sigma_{\sigma]\nu} + \theta_{\nu[\rho}\sigma_{\sigma]\mu} \,, \tag{27}$$

$$P^{(\mathbf{e}\times\mathbf{m})}_{\mu\nu\rho\sigma} = \theta_{\mu(\rho}\sigma_{\sigma)\nu} - \theta_{\nu(\rho}\sigma_{\sigma)\mu} \,. \tag{28}$$

The two first operators form the complete set of orthogonal projectors, and the two last ones mix the symmetric and the antisymmetric sectors. Then we can rewrite the total field equation using the projections as follows:

$$k^2 \left[ aP^{(\mathbf{g})}_{\mu\nu\rho\sigma} + (a - 3c)P^{(\mathbf{s})}_{\mu\nu\rho\sigma} + yP^{(\mathbf{b})}_{\mu\nu\rho\sigma} + \frac{1}{2}x \left( P^{(\mathbf{e})}_{\mu\nu\rho\sigma} + P^{(\mathbf{m})}_{\mu\nu\rho\sigma} + P^{(\mathbf{m}\times\mathbf{e})}_{\mu\nu\rho\sigma} + P^{(\mathbf{e}\times\mathbf{m})}_{\mu\nu\rho\sigma} \right) \right] \tilde{B}^{\rho\sigma} = 2\tau_{\mu\nu} \,. \tag{29}$$

Remarkably, the spin-1 parity-odd electric-momentum subspace (**m**) $\oplus$ (**e**) has always a degenerate propagator, even when the function $x$ is non-vanishing. This degeneracy reflects the propagation of an $n$-vector. The symmetry that eliminates it is

$$\tilde{B}_{\mu\nu} \to B_{\mu\nu} + \partial_\mu V_\nu \,, \tag{30}$$

whose symmetric part is the diffeomorphism and the antisymmetric part the two-form gauge redundancy. We have one iff we have the other. Interestingly, this $V^\mu$ can be identified as the Cartan's radius vector. It cannot be fully confined into any of the spin subspaces, but it corresponds to some of the components of the "momentum" and some of the components of the "electric" vector pieces from the symmetric and the antisymmetric sectors, respectively. In $n = 4$ we may understand that the 4 components of $V_\mu$ are separated into the 2+2 transverse modes of the 2 massless 3-vectors which can only be unleashed in unison. In fact, in terms of the spin projectors (omitting their indices from now on), we can rewrite $P^{(\mathbf{m}\times\mathbf{e})} + P^{(\mathbf{e}\times\mathbf{m})} = P^{(\mathbf{m})} + P^{(\mathbf{e})}$ above, apparently eliminating the degeneracy and the coupling. For these non-trivial reasons, we can invert (29) into the propagator $\Pi$ that becomes:

$$\Pi = \frac{P^{(\mathbf{g})}}{ak^2} + \frac{P^{(\mathbf{s})}}{(a - 3c)\,k^2} + \frac{P^{(\mathbf{b})}}{yk^2} + \frac{P^{(\mathbf{m})} + P^{(\mathbf{e})}}{xk^2} \,. \tag{31}$$

We have arrived at the main result of this paper.

## 4. Applications

Let us then look at the implications of (31) in a few different contexts of frame field theories.

- The **teleparallel equivalent of GR** [5,24] corresponds to $c_1 = \frac{1}{4}$, $c_2 = \frac{1}{2}$, $c_3 = -1$, and vanishing higher order terms. These imply $a = c = 1$ and $x = y = 0$. From the formulas (13) and (14) it appears that we may reproduce equivalent theories by many other choices of parameters, but it is important to note that this would require non-analytic functions of the form $c_i \sim 1/\Box$ for $i > 3$. Thus, the action of the teleparallel equivalent of GR is unique (up to irrelevant boundary terms) already at the linear order. For further convenience we define this action as $I = -\int \mathrm{d}^n x \mathrm{e}T$, introducing the torsion scalar $T \equiv T_{\alpha\mu\nu}\left(\frac{1}{4}T^{\alpha\mu\nu} + \frac{1}{2}T^{\mu\alpha\nu}\right) - T_\mu T^\mu$.

- The **modified teleparallel $f(T)$ gravity** [25,26] is given by a nonlinear function of the torsion scalar. Such models have received considerable attention in the literature [27,28], but nevertheless the nature of their degrees of freedom remains undisclosed, see e.g., [29,30] for current discussion. There is evidence [31,32] that the $f(T)$ models would, in general, contain a propagating extra

degree of freedom [33] or more [34], but here we confirm the well-known fact that in flat space the propagator reduces to that of GR. That could imply that this class of modified gravity models has a strong coupling problem. Indeed there are disturbing bifurcations in the characteristics [32] and constraint structure [33].

- The **modified teleparallel** $f(T, B)$ **gravity** [35,36], where we have the boundary term $B = \mathcal{D}_\mu T^\mu$ in terms of the metric Levi-Civita connection $\mathcal{D}_\mu$, has been motivated by the relation of the metric Ricci curvature $\mathcal{R}$ and the torsion invariants, $\mathcal{R} = -T + 2B$, due to which these models can be also considered as $f(T, \mathcal{R})$ gravity [37,38]. Indeed, we obtain the four functions as $a = f_T$, $c = f_T - f_{BB}\Box$, $x = y = 0$, implying that they propagate an extra scalar degree of freedom, in an analogy to the well-studied $f(\mathcal{R})$ models. The scalar field has the mass $\sim 1/\sqrt{f_{BB}}$, and therefore one should have $f_{BB} > 0$ to avoid a tachyonic instability.

- The **New GR** [8,11,12] was considered at the linear order in e.g., Section 4.6 of [15], and its field content can be deduced from detailed analyses in the more general context of the Poincaré gauge theory [14,16,20]. The one-parameter class of theories $2c_1 + c_2 + c_3 = 0$ i.e., $x = 0$ involves $(n^2 - 3n)/2$ components of $b_{\mu\nu}$ due to the symmetry $b_{\mu\nu} \rightarrow b_{\mu\nu} + \partial_{[\mu} v_{\nu]}$, where $v_\mu$ is an arbitrary vector. However, note that though originating from the "magnetic" pseudovector, the Kalb-Ramond field has helicity 0 since at the massless limit, oppositely to the Maxwell field, it is the longitudinal mode that remains while the transverse modes decouple.

- The generic theory (4) is a **higher-derivative New GR**. One should set $a = 1$ to obtain the canonical normalisation for the graviton. Now the gravitational wave also may possess a breathing mode, which propagates healthily, given that either $c > 1$ or $c < 1/3$. It is possible to give a mass to this scalar, but not to the graviton nor the scalar particle associated with the Kalb-Ramond field, without introducing ghosts or non-analytic functions $c_i$. Again, the crucial symmetry (30) requires $x = 0$, and the Kalb-Ramond field is not a ghost given that $y > 0$. The phenomenological viability of these models might be worth investigations.

- From the perspective **premetric teleparallelism**, the prescription (4) can be seen as the specification of the constitutive law that is reversible and linear but nonlocal. Recently a thorough analysis was performed for teleparallel gravity theory defined by a linear and local constitutive law [39]. From that analysis we can see that by setting the irreducible component "principal-1" of the constitutive law [39] to vanish gives $a = c = y = c_2$ and $x = 0$, thus eliminating precisely effects of the vector $V$ from the spectrum. Constitutive laws with infrared nonlocality have been proposed [40,41].

- The prototype **infinite-derivative gravity** [10,42,43] is given by $a = c = e^{\frac{-\Box}{M^2}}$, $x = y = 0$, where $M^2$ is the energy scale of non-locality. Such gravity theories could avoid both ghosts and singularities, and indeed they are often studied both at classical [44–46] and quantum [47] levels. We note that the teleparallel prototype theory can be realised simply as $I = -\int \mathrm{d}^n x e e^{\frac{-\Box}{M^2}} T$, whereas in the purely metric formulation the action requires the superposition of the Einstein-Hilbert and a more complicated term that is quadratic in the Riemann curvature [10].

To consider even more general frame field theories, it might be interesting to relax our main assumptions of (1) metric-compatibility (2) parity-invariance or (3) analyticity[3]. To proceed towards nonlinear orders, a natural first step would be to repeat the computation in an (a)dS background. Of course, one can also add further fields besides the frame field. We will present one interesting example, wherein we add a scalar field $\phi$ for the purpose of promoting the previous example into a scale-invariant theory.

---

[3]    Our results can be used to immediately read off the field content of the many models considered in e.g., [48].

- An example of a **scale-invariant teleparallel theory** is given (in $n = 4$ for simplicity) by

$$L = \phi^2 \left( \frac{1}{4} T^{\alpha\mu\nu} + \frac{1}{2} T^{\mu\alpha\nu} - \frac{1}{3} g^{\alpha\nu} T^{\mu} \right) e^{-\phi^2 \Box} T_{\alpha\mu\nu} - 6 \left( D_\mu \phi \right) \left( D^\mu \phi \right) , \qquad (32)$$

where the covariant derivative involves the torsion $D_\mu = \partial_\mu - \frac{1}{3} T_\mu$ [49,50]. This action is invariant under the conformal transformation of the coframe $e^a{}_\mu \rightarrow e^\theta e^a{}_\mu$ accompanied by the rescaling $\phi \rightarrow e^\theta \phi$. We can choose the gauge $\phi = 1/M$ in order to explicitly recover the previous case. It is not possible to adjust the coefficients above without either breaking the scale-invariance or the symmetry (30).

## 5. Conclusions

To summarise our derivations, we deduced that the most general quadratic torsion action contains nine free functions, and found that four of them are independent at the linear order and appear in the propagator (31). The purely metric sector of the theory is determined by the two independent functions $a$ and $c$ which describe the propagation of the graviton and the dilaton. Now there is also the function $y$ which determines the propagation of the Kalb-Ramond field, and the function $x$ which controls the non-conservation of matter energy-momentum and the propagation of the Cartan radius vector.

An issue we did not touch on in this paper was raised recently in [29]. The degrees of freedom depend upon the background geometry. Which is now the geometry that defines a physical observer? We have assumed here that fluctuations around the vacuum $\vartheta_a = \delta_a$ occur in the Weitzenböck geometry $\omega^a{}_b = 0$. It appears to be consistent as well to consider that observations take place, say, in the properly parallelised geometry $\omega^a{}_b = d\delta_b(B^a)$. The results in this prescription would be obtained from the above simply by erasing the antisymmetric fluctuations.

**Author Contributions:** Conceptualization, methodology, validation, formal analysis, investigation, writing–original draft preparation, writing–review and editing: T.K. and G.T.; supervision, project administration, funding acquisition: T.K.

**Funding:** This research was funded by the Estonian Research Council grant PRG356 and project No. MOBTT86.

**Acknowledgments:** A comment from Jose Beltrán helped to correct a mistake about $x \neq 0$ and to considerably improve the paper.

**Conflicts of Interest:** The authors declare no conflict of interest.

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
