# Peer review of "The Spectrum of Teleparallel Gravity"

_universe, doi:10.3390/universe5030080_

Round 1
Reviewer 1 Report
The submitted manuscript is an interesting and valuable extension of the work of van Nieuwenhuizen and other works in NGR by considering higher order derivative terms in the Lagrangian (3) and hence having the most general Lagrangian quadratic in torsion. They derive the propagator in this general setting and briefly discuss applications to various teleparallel theories. While I think that this discussion could have been more detailed, I consider this to be an interesting contribution and I recommend its publication in this special issue of the journal.
Two small details that I believe to be typos and that should be easily fixed:
1. Under equation (3), the authors say that 4 terms in the second line are 4th derivative while 2 terms in the 3rd line are sixth derivative. However, there are 5 and 1 terms, respectively.
2. At the end of the same paragraph, the authors say that both \omega and \Lambda are antisymmetric. \Lambda is a general LLT matrix that is not antisymmetric. When written in the infinitesimal form (4), A^a_b is antisymmetric.
Author Response
p.p1 {margin: 0.0px 0.0px 0.0px 0.0px; font: 12.0px 'Andale Mono'; color: #29f914; background-color: #000000} span.s1 {font-variant-ligatures: no-common-ligatures}
We are glad that the Reviewer 1 appreciates the manuscript as an interesting and valuable extension of previous works. We are grateful for the careful reading of the manuscript and spotting the two typos mentioned in the report. Both have been fixed in the revised version.
Reviewer 2 Report
Report on the manuscript Universe 437384, "THE SPECTRUM OF TELEPARALLEL GRAVITY"
The authors present a short manuscript in which they address the number of degrees of freedom of a large class of teleparallel theories of gravity. They
construct the Lagrangian density in a not conventional way, with quantities that are arbitrary functions of the d'Alembertian operator, including an overall
inverse d'Alembertian (i.e., the Green's function of the d'Alembertian operator) in eq. (3). These non-local quantities were introduced long ago, presumably
in ref. [23] of the manuscript, and considered also in refs. [10] and [43], at least, mainly in the context of approaches to quantum gravity. In my opinion,
this mathematical construction needs further explanation and justification. For instance, how does one obtain the field equations (16), with the inverse
d'Alembertian, from the Lagrangian (10), namely, what kind of variation leads to eq. (16)? I noticed that one of the authors wrote a paper (arxiv:0807.3778) on non-local Newtonian gravity, with few more details about the use of Green's functions.
In my opinion, the manuscript is well motivated, but the results and conclusions are not enough to justify a publication. I suggest the authors to improve the
manuscript, so that the mathematical formalism of non-local and/or infinite derivative gravity, as discussed on page 6, is better explained.
In addition, I make three remarks.
1. An arbitrary matrix of a Lorentz transformation is definitely not antisymmetric, as stated in the first line on page 3, otherwise the identity transformation is excluded (antisymmetric quantities are obtained by linearisation).
2. How does the antisymmetric tensor A arises on the right hand side of eq. (8)? Where does it come from?
3. Why the connection in eq. (1) must be purely inertial?
Author Response
p.p1 {margin: 0.0px 0.0px 0.0px 0.0px; font: 12.0px 'Andale Mono'; color: #29f914; background-color: #000000} p.p2 {margin: 0.0px 0.0px 0.0px 0.0px; font: 12.0px 'Andale Mono'; color: #29f914; background-color: #000000; min-height: 14.0px} span.s1 {font-variant-ligatures: no-common-ligatures}
We thank the referee for the useful remarks.
Unfortunately our use of the inverse d'Alembertian is confusing, since it appears in the equations only for dimensional convenience (to keep the functions c_i etc dimensionless). We have mentioned at a couple of points that we restrict to analytic operators i.e. no inverse d'Alembertians are assumed (thus, the inverse operators that appear in the equations are cancelled by the functions they multiply). This is a convention often used in the literature on ultraviolet non-local gravity. We added a the footnote 2 to clarify this better.
1. We thank the referee for spotting the typo. We have corrected it.
2. The explicit definition is included in the revision.
3. The connection is purely inertial because we restrict the present study to the frame field is the only propagating field in the theory.
Reviewer 3 Report
Please see my comments in the pdf attached.

Author Response
p.p1 {margin: 0.0px 0.0px 0.0px 0.0px; font: 12.0px 'Andale Mono'; color: #29f914; background-color: #000000} p.p2 {margin: 0.0px 0.0px 0.0px 0.0px; font: 12.0px 'Andale Mono'; color: #29f914; background-color: #000000; min-height: 14.0px} span.s1 {font-variant-ligatures: no-common-ligatures}
The reviewer 3 raises 4 points which are addressed below.
1. Ref. [1], which cites our paper, lists the 9 terms in our action in their torsion sector. Since they don't derive the general propagator there are no results to compare with ours. Refs. [2,3] are about non-analytic models which we deliberately left out from consideration. Nevertheless we added a comment and a citation to [2] in the footnote 3.
2. We have stated in the abstract the main conclusion: the viable models can include a) UV completion previously considered in the metric context, b) an extra scalar mode and c) the Kalb-Ramond field. This is again explained in the introduction. Yet, in addition to the Section 4 where the implications of the results are discussed from various perpectives, we have the brief section 5 which again summarises the main conclusions. These must be clear at this point.
Yet, we end with a hint about a possible issue that puts doubts on all the results in this paper (and all previous related works). After this final remark, all may not be clear anymore and the reader might be left in a state of doubt, even confusion. This is life, in especially in science. The issue simply isn't understood and needs to be resolved in the future.
3. We prefer not to emphasise that we have assumed teleparallel gravity, metric compatibility and flat spacetime. What we have assumed is a covariant theory where the frame field is the only propagating field, and what follows is that the action can be written solely in terms of torsion. As we have not assumed a metric in the beginning, it would be not be appropriate to state that we would have assumed metric compatibility. Flatness and metric-compatibility are properties of a connection, not of spacetime. As we define the metric to be, effectively, composed from the tetrad as usual, this metric comes with the associated Christoffel symbols as usual. The latter of course in general has curvature. Therefore it could be misleading to say that we have assumed flat spacetime. Due to such ambiguities in interpretation and use of terminology, we prefer to only emphasise that we simply study the generic frame field theory.
4. Our result is general. It applies in the mentioned special cases and they are discussed in the paper. When a propagator diverges, the corresponding field does not propagate. In case iv) this happens to the graviton, in case iii) to the Cartan vector, in case iii) to the Kalb-Ramond field and in case iv) to all metric fluctuations. Our conclusion is that iii) is a necessary requirement for any viable model.
Round 2
Reviewer 2 Report
Report on the revised manuscript Universe 437384, "THE SPECTRUM OF TELEPARALLEL GRAVITY"
The authors have not addressed the main issue presented in my first report. I presume that most readers will find that the presentation of the theory in
section 2 is too much abbreviated and lacks clarity regarding the use of the d'Alembertian operator. It seems that the use of this operator is just a trick
to avoid the introduction of constants with dimensions in the Lagrangian density. But how does one obtain the field equations (16) with an inverse
d'Alembertian operator, and what does this inverse operator mean in the field equations (16)? The variation of the Lagrangian density is not discussed at all. If one introduces constants with dimension of length, instead of inverse operator, will the field equations remain the same? I maintain that the
mathematical construction needs further explanation and justification.
Item 1 of my previous report has not been satisfactorily answered, since in the new text we read "... where both ... is antisymmetric", in the second line of
page 3.
In my opinion, the manuscript is not suitable for publication.
Author Response
There are actually no inverse operators. E.g. in Eq.(16) there is no inverse operator because since the Fourier function of the function f would start as $f_1\Box + \dots$
Since this first point remained confusing, we have added expanded the clarification further, reiterating explicitly that "The inverse operators are included only for the convenience of keeping the $c_i$ dimensionless. Thus, we assume that $c_1$, $c_2$ and $c_3$ are analytic, as well as $c_i/\Box$, where $i=4,5,6,7,8$, and $c_9/\Box^2$. Therefore all the apparent inverse d'Alembertians actually cancel from the action and the field equations as well."
We thank the referee for spotting the typo that was left after the corrections. We now eliminated it, replacing the sentence by "The spin connection $\bomega_{ab}$ is antisymmetric."
Reviewer 3 Report
The authors addressed my points so I recommend this paper for publication.
Author Response
We thank the referee again.